# Novel Flavivirus Antiviral That Targets the Host Nuclear Transport Importin α/β1 Heterodimer

**DOI:** 10.3390/cells8030281

**Published:** 2019-03-24

**Authors:** Sundy N. Y. Yang, Sarah C. Atkinson, Johanna E. Fraser, Chunxiao Wang, Belinda Maher, Noelia Roman, Jade K. Forwood, Kylie M. Wagstaff, Natalie A. Borg, David A. Jans

**Affiliations:** 1Nuclear Signaling Laboratory, Monash Biomedicine Discovery Institute and Department of Biochemistry and Molecular Biology, Monash University, Clayton, Vic. 3800, Australia; sundy.yang@monash.edu (S.N.Y.Y.); johanna.fraser@eliminatedengue.com (J.E.F.); chunxiao.wang@monash.edu (C.W.); belinda.maher@monash.edu (B.M.); kylie.wagstaff@monash.edu (K.M.W.); 2Infection & Immunity Program, Monash Biomedicine Discovery Institute and Department of Biochemistry and Molecular Biology, Monash University, Clayton, Vic. 3800, Australia; sarah.atkinson@monash.edu (S.C.A.); natalie.borg@monash.edu (N.A.B.); 3School of Biomedical Sciences, Charles Sturt University, Wagga Wagga, NSW 2650, Australia; nroman@csu.edu.au (N.R.); jforwood@csu.edu.au (J.K.F.)

**Keywords:** flavivirus, nuclear transport inhibitors, importins, viral infection, dengue virus

## Abstract

Dengue virus (DENV) threatens almost 70% of the world’s population, with no effective vaccine or therapeutic currently available. A key contributor to infection is nuclear localisation in the infected cell of DENV nonstructural protein 5 (NS5) through the action of the host importin (IMP) α/β1 proteins. Here, we used a range of microscopic, virological and biochemical/biophysical approaches to show for the first time that the small molecule GW5074 has anti-DENV action through its novel ability to inhibit NS5–IMPα/β1 interaction in vitro as well as NS5 nuclear localisation in infected cells. Strikingly, GW5074 not only inhibits IMPα binding to IMPβ1, but can dissociate preformed IMPα/β1 heterodimer, through targeting the IMPα armadillo (ARM) repeat domain to impact IMPα thermal stability and α-helicity, as shown using analytical ultracentrifugation, thermostability analysis and circular dichroism measurements. Importantly, GW5074 has strong antiviral activity at low µM concentrations against not only DENV-2, but also zika virus and West Nile virus. This work highlights DENV NS5 nuclear targeting as a viable target for anti-flaviviral therapeutics.

## 1. Introduction

Two thirds of the world population are currently at risk of dengue fever, which is an acute, mosquito-transmitted viral disease caused by dengue virus (DENV) characterized by fever, headache, severe retro-orbital pain, myalgia, rash, nausea, and vomiting [1]. Despite 96 million symptomatic cases, 500,000 hospitalizations and ca. 20,000 deaths annually, there is no approved antiviral agent currently available to treat DENV infection [1] and no efficacious DENV vaccine [2,3]. Infections can be caused by any of the four immunologically distinct DENV 1–4 serotypes, where infection with one serotype fails to provide cross-protective immunity. Infection with a second distinct DENV serotype can lead to antibody-dependent enhanced (ADE) infection, resulting in dengue haemorrhagic fever (DHF) or dengue shock syndrome, with children being particularly at risk [2]. Clearly, there is an urgent need to develop new vaccines and effective therapies.

DENV is a flavivirus closely related to West Nile virus (WNV), zika virus (ZIKV) and Japanese encephalitis virus (JEV) [4], containing a positive-sense, single-stranded RNA genome encoding three structural and seven nonstructural (NS) proteins. Highly conserved across DENV 1–4, as well as WNV/ZIKV/JEV [4,5], NS5 is essential for viral replication, through the enzymic activity of its N-terminal methyltransferase and the C-terminal RNA-dependent RNA polymerase (RdRp) domains [6]. Although DENV replication occurs in the cytoplasm, NS5 is known to shuttle between the nucleus and cytoplasm during infection through the action of specific members of the host importin (IMP) superfamily of nuclear transport proteins [7,8]. IMPs mediate signal-dependent transport of specific cargoes between cytoplasm and nucleus; importantly, the gene products of many RNA viruses, including flaviviruses, carry conserved nuclear targeting signals within their primary sequences that enable them to hijack IMPs in order to access the nucleus [5,8,9,10]. DENV NS5 is an example thereof, localising strongly in the infected cell nucleus through recognition by the IMPα/β1 heterodimer [7,11]. DENV NS5 nuclear localisation is essential for productive infection, as indicated by the fact that specific mutations in NS5 preventing IMPα/β1 recognition result in an attenuated virus, and that specific inhibitors of NS5–IMPα/β1 interaction, such as the IMPα targeting agent ivermectin, are potent anti-DENV agents [7,12,13,14]; similar results have been recently reported for WNV [15] and ZIKV [16]. The specific roles of NS5 in the host nucleus appear to be to suppress the host antiviral response [17,18], in part through impacting host mRNA splicing [19].

We previously used high-throughput screening to identify a number of novel compounds [20], including ivermectin [12,13] and *N*-(4-hydroxyphenyl) retinamide (4-HPR) [14] as potent inhibitors of DENV NS5 recognition by IMPα/β. Ivermectin was shown to be an IMPα targeting agent [13,20] with broad antiviral activity towards HIV [13], DENV [12,13] and Venezuelan equine encephalitis virus [21], whereas 4-HPR was shown to be specific for NS5 and able to inhibit all forms of DENV disease, including the ADE form [14], as well as WNV and ZIKV infection [14,16]. Here, we characterised for the first time the small molecule GW5074 as an inhibitor of the NS5–IMPα/β1 interaction that targets IMPα rather than NS5. We used a range of approaches to show that GW5074 is able to bind to the IMPα armadillo (ARM) repeat domain, leading to effects on thermal stability and *α*-helicity, and preventing binding to IMPβ1, as well as dissociating preformed IMPα/β1 heterodimer. Significantly, the results establish for the first time that GW5074 has strong antiviral activity against not only DENV-2 but also ZIKV and WNV, confirming NS5 nuclear localisation as a viable target for the development of anti-flaviviral therapeutics.

## 2. Materials and Methods

### 2.1. Inhibitors

GW5074 (G6416) was sourced from Sigma-Aldrich, St. Louis, MO, USA and *N*-(4-hydroxyphenyl) retinamide (4-HPR; 13-961-0) was from Tocris Bioscience, Bristol, UK; both were dissolved in Dimethyl sulfoxide (DMSO).

### 2.2. Cell Culture and Virus Propagation

Cells of the Vero (African green monkey kidney) and BHK-21 (baby hamster kidney) lines were maintained at 37 °C in Dulbecco’s modified eagle serum (DMEM; Sigma-Aldrich, St. Louis, MO, USA) media containing 10% heat-inactivated foetal bovine serum (FBS; Sigma-Aldrich, St. Louis, MO, USA, in a humidified incubator supplemented with 5% CO_2_ atmosphere, whereas C6/36 (*Aedes albopictus*) cells were maintained at 28 °C in Basal Medium Eagle (BME; Sigma-Aldrich, St. Louis, MO, USA) media supplemented with 10% heat-inactivated FBS [13,14]. Viral stocks of DENV-2 (New Guinea C; M29095) were propagated in C6/36 cells [14], and of ZIKV (Asian/Cook Islands/2014) and WNV (MRM61C) in Vero cells; cells at 80% confluency were infected at a multiplicity of infection (MOI) of 0.1. At 48 h, when >70% of the cells were detached, the supernatant was harvested as the virus stock. Viral titre was subsequently determined by plaque assay (see below).

### 2.3. Protein Expression

Recombinant proteins His_6_-DENV2 (TSV101) NS5 [13], His_6_-DENV3 RNA dependent RNA polymerase domain (RdRp) (Genbank accession AY662691), His_6_-ZIKV RdRp (Brazil) [16], and mouse His_6_-IMPα2 with and without the IMPβ1-binding domain (residues 67–529; IMPαΔIBB) were expressed and purified by Ni^2+^-affinity chromatography as previously [14,20]. Mouse IMPα2 and mouse IMPβ1 proteins were expressed as glutathione S-transferase (GST) fusion proteins and purified using glutathione S-beads as described [13,14,20]. Biotinylation of IMPs was carried out as previously [20].

### 2.4. AlphaScreen

AlphaScreen binding assays were performed as previously [13,14,20]. IC_50_ analysis was performed using 30 nM His_6_-IMPα binding to 5 nM biotinylated-GST-IMPβ1, and 30 nM His_6_-DENV NS5 binding to 10 nM prebound His_6_-IMPα/biotinylated-IMPβ1 heterodimer [13,14,16,20].

### 2.5. Thermostability Assay

Thermostability analysis using the fluorescent dye SYPRO orange (s6650; Thermo Fisher Scientific, Waltham, MA, USA) was performed in a Rotor-Gene Q6 plex, programmed in melt curve mode. First, 2 or 5 µM recombinant protein in phosphate-buffered saline (PBS) was mixed with or without DMSO or compounds, and then heated from 27–90 °C at a rate of 1 °C/min. Fluorescence intensity due to Sypro orange binding was measured using excitation at 530 nm and emission at 555 nm [22]. The thermal melt point (T_m_), represents the temperature at which 50% of the protein is unfolded. T_m_ was plotted against inhibitor concentrations using GraphPad Prism 7 software (GraphPad Software, San Diego, CA, USA).

### 2.6. Immunofluorescence

Vero cells were treated for 2 h without or with inhibitors prior to infection with DENV-2 at a MOI of 4, fixed 24 h later, and then stained using human anti-NS5 primary [14] and AlexaFluor 488-coupled secondary (H10120; Invitrogen, Carlsbad, CA, USA) antibodies. The 2 h treatment was found to be optimal to enable visualisation of NS5 in the infected cells. Images were captured using a Nikon C1 inverted confocal laser scanning microscopy (CLSM) in Kalman mode, with Argon laser and 100× oil immersion objective NIS Elements Software (Nikon, Tokyo, Japan). Quantitative analysis was performed on the captured images using the Image J public domain software to determine the nuclear to cytoplasmic fluorescence ratio (Fn/c) as previously [7,11,17,20].

### 2.7. Circular Dichroism Spectroscopy

Circular dichroism (CD) spectra of IMPs (0.1 mg/mL) in PBS were recorded from 190 to 250 nm in a 1 mm quartz cuvette at 20 °C using a Jasco J-815 CD spectrometer (Jasco, Easton, MD, USA). The α-helical content was calculated from the ellipticity at 222 nm as described [23]. Mean ellipticity values per residue (θ) were calculated as θ = (3300 × m × ΔA)/(lcn), where l is the path length (0.1 cm), n is the number of residues, m is the molecular mass in Daltons, and c is protein concentration in mg/mL.

### 2.8. Analytical Ultracentrifugation

Sedimentation velocity experiments were conducted using an Optima analytical ultracentrifuge (Beckman Coulter, Indianapolis, IN, USA) at a temperature of 20 °C. The IMPα/β1 heterodimer was generated by incubating equimolar concentrations of the proteins at room temperature for 30 min in IMP binding (IB) buffer containing 1 mM DTT [7,13,14] prior to centrifugation. Protein was diluted in PBS and where relevant, 12.5 or 50 μM GW5074 was added prior to centrifugation. Then, 380 μL of sample and 400 μL of reference solution (PBS) were loaded into a conventional double sector quartz cell and mounted in a Beckman 8-hole An-50 Ti rotor (Beckman Coulter, Indianapolis, IN, USA). Samples were centrifuged at a rotor speed of 40,000 rpm and the data were collected continuously at multiple wavelengths (233, 280 and 450 nm). Solvent density (1.0052 g/mL at 20 °C) and viscosity (1.0189 cP at 20 °C), as well as estimates of the partial specific volume (0.7384 mL/g for IMPα/β1 at 20 °C), were calculated using the program SEDNTERP [24,25]. Sedimentation velocity data were fitted as previously [25] to a continuous size (*c*(*s*)) distribution model using the program SEDFIT [26].

### 2.9. Cell Cytotoxicity Assay

Vero cell viability was determined by XTT (sodium 3′-[1-[(phenylamino)-carbony]-3,4-tetrazolium]-bis(4-methoxy-6-nitro) benzene-sulfonic acid hydrate) assay as previously [14,20], using XTT sodium salt (X4626; Sigma-Aldrich, St. Louis, MO, USA) and PMS (phenazine methosulfate, P9625, Sigma-Aldrich, St. Louis, MO, USA). Cells were treated for 4 h with increasing concentrations of GW5074, and XTT added 22 h later.

### 2.10. Plaque Assay

BHK-21 (used for DENV-2) or Vero (used for ZIKV/WNV) cells were seeded into 24-well plates at a density of 2 × 10^5^ cells/well and grown overnight in culture medium at 37 °C with 5% CO_2_ prior to infection. The virus inoculum was removed and replaced with semisolid overlays of 0.8% aquacide II (178515KG; Calbiochem, San Diego, CA, USA) in DMEM containing 2% FBS, and the mixture was incubated at 37 °C with 5% CO_2_. After 3–4 days, the cells were fixed with neutral buffered formalin (HT501128; Sigma-Aldrich, St. Louis, MO, USA) for 2 h at room temperature, rinsed with tap water, and stained with 1% crystal violet for 10 min. The stain was removed by rinsing the cells with tap water, and the viral plaques were counted visually. Dose–response curves were plotted using GraphPad Prism 7 software (GraphPad Software, San Diego, CA, USA).

### 2.11. Quantitative Reverse Transcription Polymerase Chain Reaction (qRT-PCR)

DENV qRT-PCR to estimate the number of viral genomes was performed as previously [14,16]. For ZIKV and WNV infections, supernatants from infected Vero cell cultures were extracted using the Isolate II RNA extraction kit (BIO-52071; Bioline Meridian Bioscience, Memphis, TN, USA), and the absolute number of RNA copies determined by TaqMan Fast Virus 1-Step Master Mix (4444432; Applied Biosystems, Foster City, CA, USA) by extrapolation from a standard curve generated from in vitro-transcribed ZIKV and WNV RNA.

### 2.12. Quantification and Statistical Analysis

Statistical parameters are defined in the Figure legends, with statistical analysis performed using GraphPad Prism 7 software (GraphPad Software, San Diego, CA, USA). In the case of Fn/c measurements (Figure 1C), n represents the number of cells measured per sample, with *p* values for significance calculated using Student’s t-test (two-tailed).

## 3. Results

### 3.1. GW5074 Inhibits DENV NS5–IMP Binding and NS5 Nuclear Localisation in Infected Cells

GW5074 was previously identified in a high throughput screen that identified the NS5 targeting agent 4-HPR [14] as an inhibitor of NS5–IMPα/β1 binding, likely through targeting IMPα/β1 rather than NS5. We set out to characterise GW5074′s properties in detail, initially confirming its ability to inhibit recognition of NS5 by IMPα/β1 (Kd of ca. 1 nM in the absence of inhibitor—Appendix A), with half maximal inhibitory concentration (IC_50_) analysis indicating an IC_50_ value of ca. 5.0 µM (Figure 1A). To confirm that GW5074 can inhibit DENV NS5 nuclear localisation in infected cells, Vero cells were treated with or without GW5074 (or the NS5 targeting agent 4-HPR as a control) 2 h prior to infection with DENV-2 (MOI 4). Immunostaining/CLSM for NS5 24 h later revealed strong nuclear localisation in the absence of inhibitor, with markedly reduced nuclear localisation/increased cytoplasmic localisation in the presence of 4-HPR, as expected [14]; treatment with 10 µM GW5074 had only a slight effect, but 20 µM GW5074 showed a marked reduction in NS5 nuclear localisation comparable to the effect of 10 μM 4-HPR (Figure 1B). Results were confirmed by quantitative analysis to determine the ratio of nuclear to cytoplasmic fluorescence (Fn/c), where NS5 nuclear accumulation in the absence of inhibitor (Fn/c of ca. 12) was reduced six-fold in 20 µM GW5074 treated cells (Fn/c of 2.5) (Figure 1C), confirming for the first time that GW5074 can inhibit DENV NS5 nuclear localisation in infected cells. That GW5074 appeared to show more activity at lower concentrations in vitro than in infected cell systems was not unexpected, taking into account the limitations of uptake and potential catabolism by living cells. Importantly, the inhibitory effects on NS5 nuclear localisation could not be attributed to cytotoxic effects of GW5074 as established in XTT assays (see Appendix A), which indicated that concentrations as high as 50 M have no impact on cell numbers/viability.

### 3.2. GW5074 Can Inhibit Binding of IMPα to IMPβ1 and Dissociate the IMPα/β1 Heterodimer

To confirm that GW5074 targets IMPs rather than NS5 [10,12], we used an AlphaScreen to assess the ability of GW5074 to inhibit recognition of SV40 large tumor antigen (T-ag) by the IMPα/β heterodimer; an IC_50_ value of ca. 4 µM was obtained (Appendix A), consistent with the idea that GW5074 targets IMPs rather than NS5 directly. We next tested whether GW5074 can inhibit binding of IMPα to IMPβ1 (Kd of ca. 6 nM; Figure 2A), showing that GW5074 could inhibit IMPα–IMPβ1 interaction (IC_50_ value of ca. 11 µM; Figure 2B, upper). Strikingly, GW5074 also appeared to be able to dissociate the preformed IMPα/β1 heterodimer (IC_50_ values of ca. 12 µM; Figure 2B, lower). The clear implication was that GW5074 is able to impact NS5 binding and nuclear transport by the IMPα/β1 heterodimer through dissociating the heterodimer itself/preventing its formation.

Sedimentation velocity analytical ultracentrifugation was used to confirm these results (Figure 2C). In the absence of GW5074, the IMPα, IMPβ1 and the IMPα/β1 heterodimer sedimented as single species with sedimentation coefficients (*s*_20,w_) of 3.5S, 5.3S and 6.7S, respectively (Figure 2C, upper and Appendix A) [26]. A clear increase in the ratio of free IMPα (3.5S) to IMPα/β1 (6.7S) was observed when the IMPα/β1 heterodimer was incubated with 12.5 μM GW5074 (Figure 2C, middle). However, at 50 μM GW5074, a ~10S species was observed, associated with a decrease in the amount of free IMPα, suggestive of dissociation of the IMPα/β1 heterodimer by GW5074, with free IMPα coexisting with an aggregated higher order complex (Figure 2C, middle). The higher order species observed when GW5074 was incubated with the IMPα/β1 heterodimer were not observed when GW5074 was incubated with IMPα alone (Appendix A), suggesting that GW5074 may not only dissociate the IMPα/β1 heterodimer, but also cause aggregation. In contrast to the results for GW5074, concentrations as high as 50 μM of the NS5 targeting agent 4-HPR, used as a control, had no effect compared to in its absence (Figure 2C, bottom). In summary, the results were consistent with the idea that GW5074 has a unique ability to dissociate the IMPα/β1 heterodimer.

### 3.3. GW5074 Binds Directly to the IMPα ARM Repeat Domain with Effects on Thermostability and Conformation

To confirm direct binding of GW5074 to IMP and explore whether the compound interacts with IMPα or IMPβ1, we exploited the spectroscopic properties of GW5074, which absorbs strongly between 350 and 500 nm, in multi-wavelength analytical ultracentrifugation experiments. We collected sedimentation velocity data at 233 nm and 450 nm; the latter enabled the specific detection of GW5074 (12.5 μM) either alone or in complex with another molecule (see wavelength scans in Figure 3A for IMPα without and with GW5074, by way of example). Overlapping *c*(*s*) distributions from analysis of data collected for IMPα and IMPα∆IBB (a truncated form of IMPα consisting of ARM repeats, lacking the autoinhibitory IMPβ-binding domain) illustrated the presence of a 3.5S species visible at 450 nm, indicative of direct binding of GW5074 to IMPα and IMPα∆IBB (Figure 3B,C). That this binding is specific to IMPα was indicated by the lack of overlap in the distributions for GW5074 and IMPβ1 (Figure 3).

Binding of GW5074 to IMPα but not IMPβ1 was further confirmed by thermostability assay [22,27,28] (Figure 4A). As shown in Figure 4A (left), IMPα, IMPαΔIBB and IMPβ1 all showed maximal thermostability between 42 and 46 °C. Strikingly, the thermostability of IMPα and IMPαΔIBB, but not IMPβ1, was markedly increased by GW5074 concentrations of 20–40 M, with maximal stability attained at >70 °C (Figure 4A, left), consistent with the idea that GW5074 binds directly to IMPα/IMPα∆IBB, and impacts conformation in doing so (Figure 4A). That the conformational changes within IMPα/IMPα∆IBB were observed at GW5074 concentrations as low as 20 μM is consistent with the results for inhibition of NS5 nuclear import (see Figure 1C), and comparable to the GW5074 concentrations leading to maximal dissociation of the IMPα/β1 heterodimer (Figure 2 and Table 1), implying that the GW5074-induced conformational effects on IMPα are the likely mechanism of inhibition of NS5 nuclear accumulation. Underlining the specificity of the results, no effects were observed for the thermostability of ZIKV or DENV3 RdRp in the presence of GW5074, consistent with lack of binding to these proteins (Figure 4A, middle). The NS5 specific agent 4-HPR, used as a negative control, showed no effect on the thermostability of IMPα∆IBB (Figure 4A, right), consistent with lack of binding to IMPα; this was in contrast to its effect on DENV3 RdRp, where concentrations >40 μM lowered thermostability from ca. 42 °C to <28 °C. The clear implication is that, in binding IMPα directly, GW5074 marked effects on IMPα conformation.

Far-UV circular dichroism (CD) spectroscopy was performed to confirm the structural changes of inhibitor binding to IMPα and IMPαΔIBB. The CD spectra for IMPα and IMPαΔIBB displayed double minima at 208 and 222 nm (Appendix A), consistent with IMPα’s predominantly α-helical structure [29], with IMPβ1 showing comparable spectra. GW5074 reduced the α-helical content of both IMPα and IMPαΔIBB (by ca. 20% and 30%, respectively), but had no effect on IMPβ1 (Figure 4B), consistent with the idea that GW5074 binding to IMPα alters folding/conformation. The results overall are consistent with the idea that GW5074 binds to the ARM repeat domain of IMPα to alter structure/conformation, which would appear to be the basis of inhibited binding to IMPβ1.

### 3.4. GW5074 Is A Potent Anti-Flavivirus Agent

To confirm the physiological relevance of the above observations, we tested the ability of GW5074 to inhibit flavivirus infection. Vero cells were infected at a MOI of 1 with DENV-2 (New Guinea C; M29095), ZIKV (Asian/Cook Islands/2014) or WNV (Kunjin; MRM61C strain) [12,13,14,15,16], followed 2 h later by the addition of increasing concentrations of GW5074. Twenty-two hours later, virus production was quantified by plaque assays and RT-qPCR analysis on the cell supernatant (see Materials and Methods). Results clearly show that GW5074 is a potent inhibitor of DENV, ZIKV and WNV infection, with EC_50_s of ca. 7 µM or less (Figure 5), whether estimations were made in terms of infectious virus (plaque-forming units; Table 2) or virus replication as estimated by RT-qPCR (Table 2). As indicated above, these effects were not attributable to cytotoxic effects of GW5074 (Appendix A), i.e., the concentrations of GW5074 effective at inhibiting flavivirus are not toxic. The higher EC_50_ observed for GW5074 inhibition of infectious virus production/virus replication for WNV compared to DENV and ZIKV presumably relates to the fact that, although NS5 is homologous in all three flaviviruses, the structure of WNV RdRp appears to be more similar to that of the flaviviridae members hepatitis C and bovine pestivirus [30] than those of DENV or ZIKV.

## 4. Discussion and Conclusions

With the continued rise in DENV infections worldwide, antivirals to combat DENV are urgently needed. This study is the first to document GW5074′s ability to inhibit flavivirus infection through limiting NS5 nuclear targeting; this adds to a growing body of literature documenting the viability of targeting viral protein nuclear import as a means to inhibit a range of viruses [8,12,13,14,15,16,20,21,25,31,32,33]. It is clear from the present study that inhibitors preventing nuclear import of key viral proteins such as DENV NS5, either by targeting IMPs directly [12,13,20,21,25] as here in the case of GW5074 or the viral component of the host–pathogen interface in the case of 4-HPR [14,31], can be efficacious in limiting flavivirus infection. Figure 6 highlights the different mechanisms of GW5074 and the known anti-DENV agent 4-HPR [14] in blocking NS5–IMPα/β1 interaction, and thereby preventing NS5 nuclear translocation and its subsequent impact on the antiviral response/host mRNA splicing, etc. [7,17,18,19,33]. Although not shown in the figure, the inhibitor ivermectin, similar to GW5074, inhibits NS5–IMPα/β1 interaction [12,13,20], and displays potent anti-DENV activity (EC_50_ of 0.4 M). Since ivermectin is structurally quite distinct from GW5074, it would be interesting to determine if it possesses a similar mechanism in impacting IMPα/β1 interaction. It would also be important to investigate aspects such as IMPα isoform specificity of both GW5074 and ivermectin; since both limit flavivirus infection through inhibition of IMPα/β1-recognition of multiple nuclear import cargoes (e.g., SV40 large T-antigen [13,20]), it seems likely that both target/inhibit more than one IMPα, but of course this would require experimental verification. In this context, the other flaviviral protein (capsid) reported to access the nucleus in infected cells for DENV and WNV also appears to be recognised by IMPα [34,35], implying that GW5074 (and likely ivermectin) should also inhibit capsid nuclear localisation, although this remains to be confirmed experimentally.

GW5074, previously known for its anti-picornaviral [32] and anti-bacterial activities [36], is shown here for the first time to possess activity against the flaviviruses DENV, ZIKV and WNV. GW5074 targets the host nuclear transport protein IMPα, eliciting conformational effects to prevent IMPα heterodimerisation with IMPβ1, and thereby blocking its ability to transport NS5 to the nucleus. The structural properties of IMPα, central to its function in nuclear transport, make it a unique target. Agents perturbing this such as GW5074, as we showed here for the first time, can exert powerful effects on IMPα cargoes, ultimately having antiviral effects in the case of DENV, ZIKV and WNV, through limiting NS5 nuclear localisation, which is so critical to the flavivirus life cycle [7,12,13,14,15,16], presumably through limiting the host antiviral response at the level of transcription (e.g., of interleukin 8 [7,17,18,33]) and mRNA splicing [19].

The fact that GW5074 binding to the ARM repeat domain of IMPα not only inhibits IMPα recognition of nuclear targeting sequences through the ARM domain, but also binding to IMPβ1 mediated by the distinct N-terminal IBB domain of IMPα, is intriguing. Since GW5074 binding impacts IMPαIBB thermostability and α-helicity, it is likely that GW5074 binding disrupts IMPα’s multiple protein–protein interactions by limiting the inherent flexibility of the ARM repeat domain; protein flexibility [37] is a key factor in enabling proteins such as IMPβ1 [38] to bind to multiple binding partners, including diverse transport cargoes, as well as IMPα through its IBB domain, RanGTP and the hydrophobic repeats of the nucleoporins that make up the nuclear pore. Since a many DNA as well as RNA viruses (e.g., Rabies, Venezuelan equine encephalitis virus, Hendra, respiratory syncytial virus, and influenza) rely on IMPα/β1 for robust infection see [8,9,39], an exciting possibility is that GW5074 may have broad antiviral activity extending to most if not all of these.

In conclusion, the results here for GW5074 underline the viability of targeting viral protein nuclear transport to have an antiviral effect. In particular, our study highlights IMPα, and the IMPα/β1 binding interface, as robust candidate targets for the development of antivirals, which are urgently required for flaviviruses such as DENV, ZIKV and WNV, as well as RNA viruses in general. Detailed crystallographic structures of inhibitors with IMPα loom as an important priority to inform future therapeutic development to derive agents that can inhibit viral protein nuclear import, but have only limited impact on host function and circumvent issues of drug resistance. This remains a focus of future work in this laboratory.

## Figures and Tables

**Figure 1 cells-08-00281-f001:**
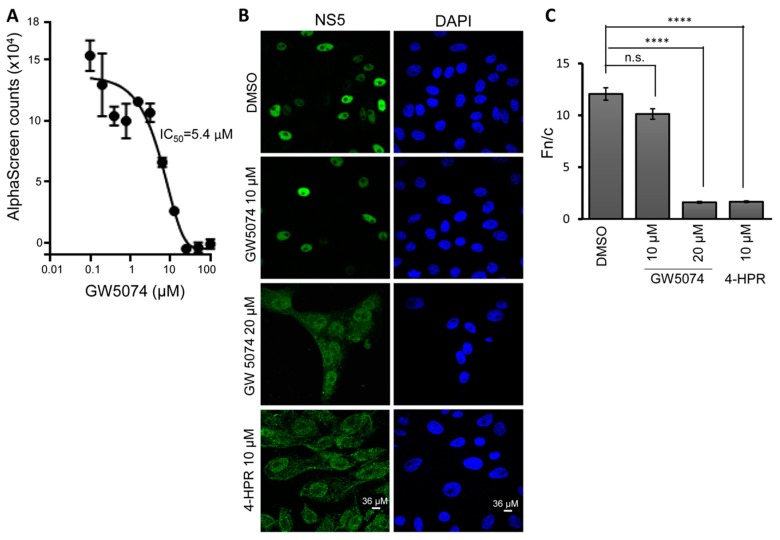
GW5074 inhibits DENV NS5 recognition by IMPα/β1 and nuclear localisation. (**A**) AlphaScreen technology was used to determine the IC_50_ for inhibition by GW5074 of binding of IMPα/β1 (10 nM) to His_6_-DENV NS5 (30 nM). Data represent the mean ± standard deviation (SD) for triplicate wells from a single experiment, from a series of two independent experiments (see Table 1 for pooled data). (**B**,**C**) Vero cells were treated with GW5074 or 4-HPR 2 h prior to infection with DENV-2 (MOI 4), followed by replacement of the medium with fresh DMEM containing 2% FCS. Cells were fixed 24 h later, stained for NS5 and imaged as described in Material and Methods, with NS5 (left) and DAPI (nuclei, right) staining indicated. (**C**) Image analysis was performed on images such as those in (**B**) using the Image J public domain software to determine the nuclear to cytoplasmic fluorescence ratio (Fn/c), where separate measurements for each cell (n > 50) were performed for the nuclear, cytoplasmic and background fluorescence. Results represent a single experiment from a series of two independent experiments. n.s., not significant; **** *p* < 0.001 (Students t-test).

**Figure 2 cells-08-00281-f002:**
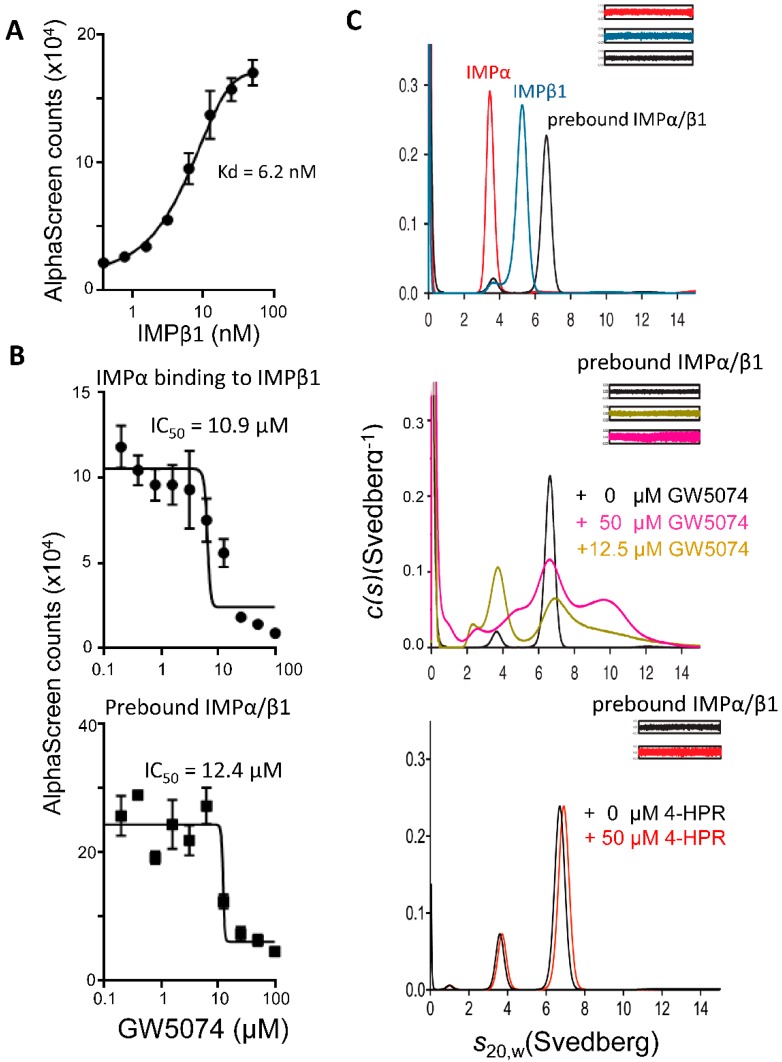
GW5074 inhibits binding of IMPα to IMPβ1 and can dissociate the IMPα/β1 heterodimer. (**A**) AlphaScreen technology was used to determine the dissociation constant (Kd) of binding His_6_-IMPα (30 nM) to biotinylated-GST-IMPβ1. (**B**) AlphaScreen technology was used to determine the IC_50_ values for inhibition of IMPα binding to IMPβ1 as well as for dissociation of prebound IMPα/β1 heterodimer for GW5074. Data (**A**,**B**) represent the mean ± SD for triplicate wells from a single typical experiment, from a series of two independent experiments (see Table 1 for pooled data). (**C**) Results from sedimentation velocity analytical ultracentrifugation experiments on purified recombinant IMPs in the absence or presence of GW5074. The continuous sedimentation coefficient distribution (*c*(*s*)) was plotted as a function of *s*_20,w_ for IMPα, IMPβ1, and IMPα/β1 in the absence of inhibitors (top). *c*(*s*) analysis of IMPα/β1 in the presence of GW5074 (middle) or 4-HPR (bottom) was also performed. Residual plots are shown in insets. Data are from a single experiment, representative of two independent experiments.

**Figure 3 cells-08-00281-f003:**
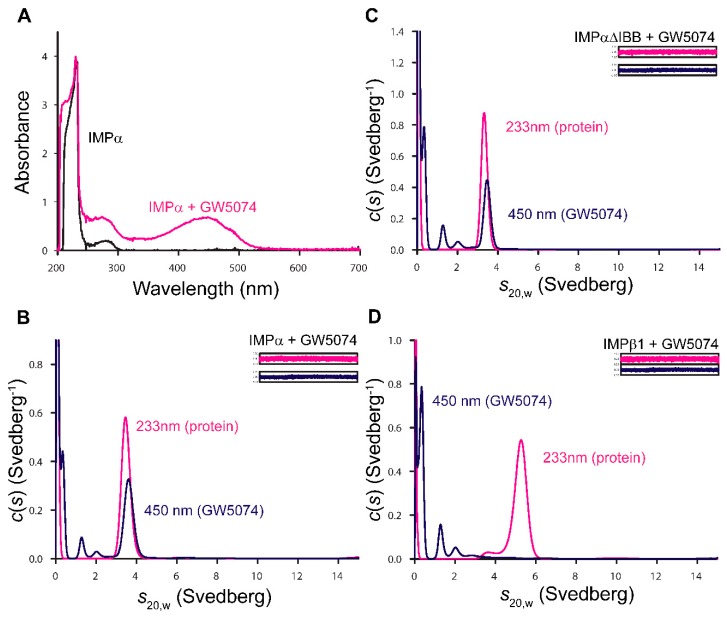
GW5074 binds directly to the IMPα ARM repeat domain as indicated by analytical ultracentrifugation. (**A**) Representative wavelength scans of 200–700 nm collected in the analytical ultracentrifuge on purified recombinant IMPα alone and in the presence of 12.5 μM GW5074. Analogous scans were obtained for IMPα∆IBB and IMPβ1 in the absence or presence of GW5074 (data not shown). (**B**–**D**) *c*(*s*) distributions from fits to data collected at 233 and 450 nm were plotted as a function of *s*_20,w_ for IMPα, IMPα∆IBB and IMPβ1 in the presence of 12.5 μM GW5074. The residual plots are shown in insets. Data are from a single experiment, representative of two independent experiments.

**Figure 4 cells-08-00281-f004:**
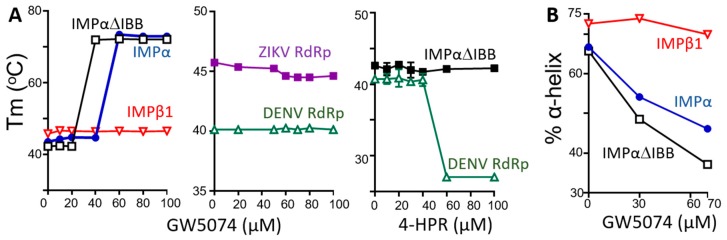
GW5074 binding to the IMPα ARM repeat domain impacts thermostability and α-helicity. (**A**) His_6_-tagged IMP proteins were subjected to thermostability analysis, as described in Section 2, in the absence and presence of increasing concentrations of GW5074 or 4-HPR to determine the Tm (°C). (**B**) The α-helical content of IMPα, IMPαΔIBB and IMPβ1 was analysed by CD spectroscopy in the presence of increasing concentrations of GW5074. α-helical content was calculated from MRE data at 222 nm, as described in Section 2 (CD spectra are shown in Appendix A). The results (**A**,**B**) are from of a single experiment, representative of two independent experiments.

**Figure 5 cells-08-00281-f005:**
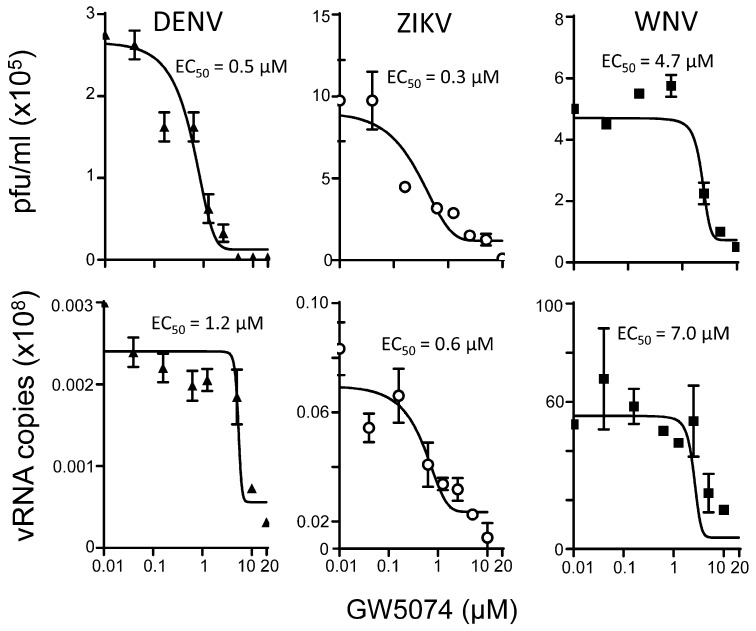
GW5074 is a potent anti-flavivirus agent. Vero cells were infected with DENV, ZIKV or WNV (Kunjin) at MOI 1 for 2 h, after which virus was removed, and fresh medium containing 2% FCS containing the indicated concentration of GW5074. Culture medium was collected 22 h later and viral titres determined by plaque assay or RT-qPCR [12,13,14,15,16]. Results represent the mean ± SD for duplicate wells from a single assay, representative of >two independent experiments. See Table 2 for pooled data.

**Figure 6 cells-08-00281-f006:**
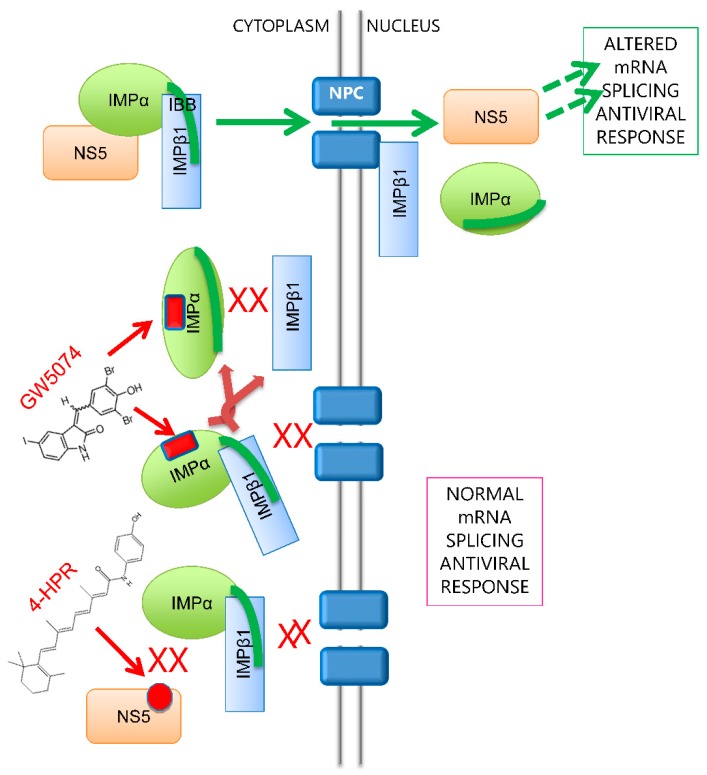
Mode of action of inhibitors blocking IMP-NS5 nuclear import. In the absence of inhibitors, NS5 is recognised by the IMPα/β1 heterodimer, where IMPα interacts with IMPβ1 via the IBB domain of IMPα (green line), followed by nuclear import to impact host antiviral responses. The IMPα targeting compound GW5074 appears to bind to IMPα to prevent it binding to IMPβ1 and hence blocks NS5 nuclear import. In contrast, the NS5 targeting compound 4-HPR prevents recognition of NS5 by IMPα/β1 heterodimer, and hence blocks NS5 nuclear import. Antiviral action of the inhibitors is through blocking NS5′s impact on mRNA splicing/the host antiviral response.

**Table 1 cells-08-00281-t001:** Summary of IC_50_ data for GW5074 from AlphaScreen analysis.

Binding Parameters		
Proteins	Kd * (nM)	IC_50_ for GW5074 * (µM)
IMPα + IMPβ1 ^a^	5.4 ± 1.1 (2) ^b^	10.3 ± 0.8 (2) ^b^
IMPα/β1 ^c^	N/A ^d^	12.5 ± 0.1 (2) ^b^
IMPα/β1: DENV NS5 ^e^	0.8 ± 0.2 (2) ^b^	6.3 ± 0.9 (2) ^b^

* Results represent the mean ± SD (n) from analysis as per Figure 1 and Figure 2. ^a^ IMPα+β1, IMPα binding to IMPβ1. ^b^ Number of independent experiments. ^c^ IMPα/β1, prebound IMPα/β1 heterodimer; ^d^ N/A, not applicable. ^e^ IMPα/β1 + DENV NS5, prebound IMPα/β1 heterodimer binding to DENV NS5.

**Table 2 cells-08-00281-t002:** Summary of EC_50_ data for GW5074 from plaque assay and RT-qPCR.

EC_50_ (µM) *			
Virus	DENV	ZIKV	WNV
Plaque assay	0.8 ± 0.4 (2) ^a^	0.3 (1) ^a^	5.2 ± 0.7 (2) ^a^
RT-qPCR	1.4 ± 0.2 (2) ^a^	0.5 ± 0.1 (2) ^a^	4.8 ± 3.8 (2) ^a^

* Results represent the mean ± SD (n) from analysis as per Figure 5. ^a^ Number of independent experiments.

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
