# Peer review of "Novel Flavivirus Antiviral That Targets the Host Nuclear Transport Importin α/β1 Heterodimer"

_cells, 2019, doi:10.3390/cells8030281_

Round 1

Reviewer 1 Report

In this article, the authors describe the ability  of the small molecule GW55074 to inhibit flavivirus infectious cycle. They show that DENV NS5 nuclear localization in infected cells is affected upon treatment with GW5074, and demonstrate that GW5074 directly interacts with importin protein IMP? through its armadillo repeat domain, leading to a conformational change that prevents IMP?/?1 heterodimer association, altering NS5 nuclear localization.

General comments:

The paper is well written, and the subject is indeed of importance as there is no antiviral treatment offered against flaviviral infections, in particular dengue and Zika virus. As mentioned by the authors, specific inhibitors of NS5-IMP?/?1 already exist such as ivermectin, that are proved to be anti-flaviviral agents. However the authors offer a demonstration of GW5074 mode of action that specifically targets IMP?/?1 heterodimer association. While ivermectin mechanism of action has not been elucidated yet, it displays comparable sub-micromolar EC50 values for DENV as that of GW5074, therefore a discussion of GW5074 effects versus those of ivermectin would be a real plus in the discussion section.

Of interest would also be to mention whether other IMPs are targeted or not by GW5074, and if the C capsid flaviviral protein, also known to display a nuclear localization, is affected by the treatment.

Specific comments:

Introduction:

- line 53 : define the abbreviation for WNV

- Please remove irrelevant text, lines 80 to 87.

- A quick description and role of NLS in general, and their presence in some flaviviral proteins, should be added in the introduction.

- material and methods :

- Please add the catalog numbers for all the products that were used.

- Figure 1:

- Have other times than 2h been examined?

- The GW5074 IC50 is 5?M for the His-tagged NS5, but 20?M of GW5074 are necessary to observe an effect with the virus itself. Please discuss this.

- Cytotoxicity test (Figure S5) should be placed with Figure 1 instead of Figure 5.

- error bar is missing for 50?M .

- line 178 : please correct the reference (14)

- line 190: (Fn/c of c.12) please explain c. or suppress

- Figure 3 :

- Please indicate the concentration of GW5074 that was used in this experiment

- Figure 4:

- There are no top and bottom panels (legend).

- Concentrations of GW5074 that impact IMP?/?1 are >40 ?M, largely above that which is necessary to observe an effect on the NS5 location (Figure 1). Thus even though GW5074 impacts this interaction, can it be inferred that it is the reason why NS5 does not localize in the nucleus anymore?

- IMP? thermostability is affected in presence of GW5074 even if ARM domains were removed.  Please discuss this.

- Figure 5:

- Could the higher EC50 with WNV be explained by differences in the NS5 structure of the different flaviviruses studied? Please discuss.

Author Response

 We would like to thank the Reviewers for their constructive suggestions, all of which have been taken into consideration in the revised manuscript. All of the Reviewers’ comments have been addressed, the necessary additions, modifications or deletions made, and the paper proofread by two native English speakers. We are convinced that the manuscript is much more complete, rigorous and clear as a result. The point-by-point responses to the comments of the Reviewers are listed below. 

Reviewer#1
     The paper is well written, and the subject is indeed of importance as there is no antiviral treatment offered against flaviviral infections, in particular dengue and Zika virus. As mentioned by the authors, specific inhibitors of NS5-IMP?/?1 already exist such as ivermectin, that are proved to be anti-flaviviral agents. However, the authors offer a demonstration of GW5074 mode of action that specifically targets IMP?/?1 heterodimer association. While ivermectin mechanism of action has not been elucidated yet, it displays comparable sub-micromolar EC50 values for DENV as that of GW5074, therefore a discussion of GW5074 effects versus those of ivermectin would be a real plus in the discussion section.

    Of interest would also be to mention whether other IMPs are targeted or not by GW5074, and if the C capsid flaviviral protein, also known to display a nuclear localization, is affected by the treatment.

To satisfy the Reviewer, we now compare GW5074’s properties to those of ivermectin in the Discussion (lines 361 to 368); as for ivermectin, the specificity of GW5074 for different IMPas has not been examined, and we now stress this in the Discussion (lines 364-368). Since GW5074 inhibits recognition by IMPa/b1of multiple nuclear import cargoes (such as SV40 T-antigen – Fig. S3) in addition to DENV NS5, it is likely that nuclear import of capsid will also be inhibited. This has now been added to the Discussion (lines 368-371) to satisfy the Reviewer.  

line 53: define the abbreviation for WNV

The abbreviation has now been defined (line 52). 

Please remove irrelevant text, lines 80 to 87.

The irrelevant text is now removed – we thank the Reviewer for pointing this out !! 

A quick description and role of NLS in general, and their presence in some flaviviral proteins, should be added in the introduction.

A short description of the role of nuclear targeting signals in importin-mediated nuclear transport, and their presence in flaviviral proteins is now included in the introduction section, lines 59 to 62.

Please add the catalogue numbers for all the products that were used.

Catalogue numbers have been added for all the products used in the Materials and Methods section. 

Figure 1: 

Have other times than 2h been examined?

2 h pretreatment with inhibitors was the optimal time point to visualise NS5 nuclear localisation. We have added this point to Section 2.6, lines 125-129. 

The GW5074 IC50 is 5?M for the His-tagged NS5, but 20?M of GW5074 are necessary to observe an effect with the virus itself. Please discuss this.

To satisfy the Reviewer, the not unexpected differences in potency of GW5074 in vitroand in vivois now discussed in the Results section, lines 199-201. 

Cytotoxicity test (Figure S5) should be placed with Figure 1 instead of Figure 5.

Figure S5 is now Figure S2, as requested, and first cited in the text in the Results section, lines 201-203.

error bar is missing for 50?M (Figure S5).

In fact the error bar is present (SD of 0.039), but hard to see because it is so small ! 

line 178: please correct the reference (14)

Corrected.

line 190: (Fn/c of c.12) please explain c. or suppress

Now defined in the text when it is used for the first time (Introduction, line 44). 

Figure 3: 

Please indicate the concentration of GW5074 that was used in this experiment.

The concentration of GW5074 was stated in Figure 3 legend (line 312) but has now been added to the Results section (line 246). 

Figure 4:

There are no top and bottom panels (legend).

Corrected.

Concentrations of GW5074 that impact IMP?/?1 are >40 ?M, largely above that which is necessary to observe an effect on the NS5 location (Figure 1). Thus even though GW5074 impacts this interaction, can it be inferred that it is the reason why NS5 does not localize in the nucleus anymore?

Even though different assays/distinct systems have distinct sensitivities and measure different outcomes, it is striking that the concentration of 20 mM that impacts nuclear localisation (Fig. 1C) is so close to the concentrations (20-40 mM) implicated by the thermostability/CD studies. We thank the Reviewer for alerting us to this, and now discuss this in the Results section (line 285-290) to satisfy the Reviewer.

IMP?thermostability is affected in presence of GW5074 even if ARM domains were removed.  Please discuss this.

The Reviewer has misunderstood – ARM domains have not been removed. ARM domains are the site of GW5074 binding (see Section 3.3; lines 303-305 summarise the findings). 

Figure 5:

Could the higher EC50 with WNV be explained by differences in the NS5 structure of the different flaviviruses studied? Please discuss.

The higher EC50 is indeed likely due to structural differences, and we have added this point to the Results section (lines 333-337) with new Reference 30 included - we thank the Reviewer !

Reviewer 2 Report

In this paper the authors describe that the small molecule GW5074 has anti-DENV action through its novel ability to inhibit NS5-IMPα/β1 interaction in vitro as well as NS5 nuclear localisation in infected cells. In addition to DENV, GW5074 also acts against ZIKV and WNV. They used a range of microscopic, virological and biochemical/biophysical approaches.

The concept of this research is interesting, however, the paper lacks robustness – only one experiment set is not enough. There is no statistical analysis described in the paper.  More detailed labeling of their samples and statistical analysis of the samples has to be included.

Several issues need to be addressed and the paper has to be proofread by English language editor.

Reviewer's comments:

General comments:

Ln 78: … for the development of anti-flaviviral. – please finalize the sentence appropriatelly

Lns 80-88: these are authors instructions…

Lns 91- :Detailed description of the compound GW5074 should be provided.

Lns 95-96: Growth medium for Vero cells should be more precisely specified, not only DMEM..?

Lns 128 - : please provide the description of the staining. Why MOI 4? How about MOI 1 and 0.1? Why did you use different MOIs?

Lns in the Methods section: the description of CLSM is missing.

Lns 184: remove “confocal laser scanning microscopy (CLSM)”

Figure 1: The staining of the nuclei or image under transmitted light would be helpful to the readers.

Figure 1: Statistical analysis is missing. In addition, how are the data presented mean +/- SEM or SD? Why differently?

Methods: Description of statistical analysis is missing.

Methods: Description of antibodies is missing.

Figures 1 - 5: The data should be pooled from at least two independent experiments.

Author Response

Reviewer#2

     In this paper the authors describe that the small molecule GW5074 has anti-DENV action through its novel ability to inhibit NS5-IMPα/β1 interaction in vitro as well as NS5 nuclear localisation in infected cells. In addition to DENV, GW5074 also acts against ZIKV and WNV. They used a range of microscopic, virological and biochemical/biophysical approaches.

     The concept of this research is interesting; however, the paper lacks robustness – only one experiment set is not enough. There is no statistical analysis described in the paper.  More detailed labeling of their samples and statistical analysis of the samples has to be included.

All experiments have been performed at least twice. This has now been spelled out in each of the figure legends. Details of the statistical analysis have also been added to the manuscript, including the Figures, and Section 2.12 added to describe the statistical tests. We thank the Reviewer for pointing out this omission. 

The paper has to be proofread by English language editor.

We have proofread the manuscript by two native English speakers, as requested.  

Ln 78: … for the development of anti-flaviviral. – please finalize the sentence appropriately

The sentence is now completed. 

Lns 80-88: these are authors instructions

Removed

Lns 91: Detailed description of the compound GW5074 should be provided.

Catalogue number added to Materials and Methods (line 87).  

Lns 95-96: Growth medium for Vero cells should be more precisely specified, not only DMEM?

Now precisely specified (lines 92-94) – we thank the Reviewer. 

Lns 128:please provide the description of the staining. 

Adescription of the staining is now included (Section 2.6, lines 125-127).  

Lns 128: Why MOI 4? How about MOI 1 and 0.1? Why did you use different MOIs?

MOI 4 is used for immunofluorescence to enable visualisation of NS5. MOI 1 is used for all other experiments, including infectious virus production. MOI 0.1 is the standard MOI for virus propagation. These experimental details are spelled out in Sections 2.2, 2.6 etc. 

Lns in the Methods section: the description of CLSM is missing

Description of CLSM imaging/imaging analysis is included in Section 2.6 (lines 128-131).  

Lns 184: remove “confocal laser scanning microscopy (CLSM)”

Removed. 

Figure 1: The staining of the nuclei or image under transmitted light would be helpful to the readers.

Images of DAPI staining (right panels) are now included in new Figure 1B as requested. 

Figure 1: Statistical analysis is missing. In addition, how are the data presented mean +/- SEM or SD? Why differently?

Statistical analysis is now included in the manuscript, as appropriate, in every figure legend.The data represent mean +/- SD, not SEM, throughout. 

Methods: Description of statistical analysis is missing.

Statistical analysis is now described in Section 2.12.

Methods: Description of antibodies is missing.

Antibodies are now detailed in Section 2.6, lines 126-127. 

Figures 1 - 5: The data should be pooled from at least two independent experiments.

The Reviewer has overlooked Tables 1 and 2, which encapsulate the pooled data (2 independent experiments) for all of the experimental data.

Round 2

Reviewer 2 Report

In this paper the authors describe that the small molecule GW5074 has anti-DENV action through its novel ability to inhibit NS5-IMPα/β1 interaction in vitro as well as NS5 nuclear localisation in infected cells. In addition to DENV, GW5074 also acts against ZIKV and WNV. They used a range of microscopic, virological and biochemical/biophysical approaches.

     The concept of this research is interesting.

In the description of CLSM - please add also the objectives, filters, laser.

Tables - please describe in the figure legend that number in brackets represent number of different experiments.

Author Response

In the description of CLSM - please add also the objectives, filters, laser.

We have added this information to the Methods section.

Tables - please describe in the figure legend that number in brackets represent number of different experiments.

We have added this information to the Tables.

We thank the Reviewer.

DAJ